# Common Home Remedies Do Not Deter Argentine Ants, *Linepithema humile* (Hymenoptera: Formicidae), from a Preferred Harborage

**DOI:** 10.3390/insects15100768

**Published:** 2024-10-04

**Authors:** Jacob B. Holloway, Daniel R. Suiter, Jerry W. Davis, Wayne A. Gardner

**Affiliations:** 1Department of Entomology, University of Georgia Griffin Camus, 1109 Experiment Street, Griffin, GA 30223, USA; jbh301@gmail.com (J.B.H.);; 2Experimental Statistics, University of Georgia, College of Agricultural and Environmental Sciences, 1109 Experiment Street, Griffin, GA 30223, USA

**Keywords:** Argentine ant, *Linepithema humile*, control, natural products, deterrence, placebo

## Abstract

**Simple Summary:**

The Argentine ant, *Linepithema humile* (Mayr), was introduced into the United States from South America in the late 1800s. Since its establishment in California and the southeastern U.S., it has become an agricultural pest by interfering with citrus production in California, but it is widely known in most of its range as a major nuisance pest. It commonly establishes nest sites in mulch and leaf litter, where colonies may be populated by hundreds of thousands of worker ants (or more) with a large home range. When searching for food, ants may discover and recruit to sweet foods inside of structures. To alleviate ant pest problems, the structural pest control industry may incorporate EPA-approved baits and residual products that are designed to reduce the ant population. Homeowners experiencing nuisance pests are often motivated to solve their own problem, and they may be susceptible to outlandish, fact-free claims of do-it-yourself “secret” solutions that “they don’t want you to know about”. In this study, we evaluated several of these recommendations. In two trials designed to evaluate a product’s ability to deter Argentine ants from a preferred nest site, we evaluated several common home remedies, namely the use of tansy plant leaves, cucumber peels, and soybean extract. None of these home remedies deterred ants from a preferred harborage. Fresh leaves from rosemary and spearmint plants, however, did deter ants, as do many commercially available essential oils.

**Abstract:**

In two laboratory trials, natural products, including freshly picked leaves from spearmint, rosemary, and tansy plants, a water extract from soybean plants, peels from a common cucumber, and 1% peppermint oil in hexane, were placed in a moist harborage preferred by Argentine ants, *Linepithema humile* (Mayr), and the number of ants entering the harborage after two and four hours was counted. None of the recommended home remedies (tansy, cucumber, or soybean extract) deterred ants from an attractive, moist harborage in either trial, even when the quantity of these treatments was increased 4- to 10-fold. Freshly picked leaves from rosemary and spearmint plants deterred ants from harboring, and the 1% peppermint oil was the most deterrent of all treatments.

## 1. Introduction

The Argentine ant, *Linepithema humile* (Mayr), is a major nuisance pest. It is a highly invasive, small (3 mm), monomorphic species from South America that forms super-colonies, and it reportedly came to the United States (U.S.) in coffee shipments into the port of New Orleans, Louisiana [1]. Concerns over the environmental impact of chemically based insecticides used to control pests, especially ants, in non-agricultural environments have led to changes in product labels that have served to limit the use of chemical insecticides for Argentine ant control. Alternative and low-impact methods used to control ants were recently reviewed [2]. In 2009, the U.S. Environmental Protection Agency (E.P.A.), in an effort to reduce non-target exposure to pyrethroid and pyrethrin products, implemented an initiative to revise guidelines for products used in non-agricultural, outdoor settings [3]. This initiative specifically limited non-agricultural outdoor uses of pyrethroids and pyrethrins to spot or crack-and-crevice treatments. In light of this change, alternative methods to conventional ant control have received attention as potential management options, with particular interest in the use of natural products, including essential oils, as deterrents and insecticides [2,4,5,6,7,8], Argentine ant semiochemicals as stand-alone applications and in conjunction with other products [2,9,10,11,12], and other less conventional products, including plants, detergents, and food items [13,14,15,16,17]; Bader [13], for example, promoted the benefits and efficacy of products that have not been investigated. In particular, Bader [13] claimed numerous plant-based products to be effective as deterrents to ant foraging. Included are claims about cucumber peels, a soybean water extract, and tansy leaves as deterrents to ants; we also evaluated the deterrent nature of freshly-harvested leafy material from spearmint and rosemary plants, as commercially available essential oil extracts from these plants are known deterrents [6,7]. Although no scientific evidence supports these claims, given the appeal of utilizing alternatives to conventional insecticides, this presented an opportunity for further investigation. The objective, therefore, of this study was to assess claims made by Bader [13] about the use of several nonconventional products as deterrents to Argentine ant harboring.

## 2. Materials and Methods

### 2.1. Ants

The Argentine ants used in our bioassays were collected from Barnesville, Georgia (N 33° 3′17.11″, W 84° 9′59.16″). The ants, including brood and queens, were collected along with accompanying soil, leaf litter, and other debris and placed in a plastic tub (≈57 × 45 × 13 cm) (Model 400-5N, Del-Tec/Panel Controls Corporation, Greenville, SC, USA). The tub was prepared in advance by coating the inside walls with Fluon^TM^ (Northern Products Inc., Woonsocket, RI, USA) to prevent ant escape. To separate ants from leaf litter debris, a moistened harborage (described below) was placed in each corner of the tub. After several days, as the leaf litter dried, the ants (workers, brood, and queens) migrated from leaf litter debris into clean, debris-free harborages that allowed for ease of use in future bioassays. Argentine ants are susceptible to desiccation, and they readily move from dry/drying habitats to moist habitats [18]. The ants were held at ambient humidity and temperature (20–23 °C), and they were provided, ad libitum, water, 25% sugar water, and freshly killed (via freezing) crickets.

### 2.2. Ant Harborages

The harborages consisted of polystyrene culture dishes (100 × 25 mm; NalgeNunc International, Rochester, NY, USA) half-filled with Castone^TM^ (Model 99044, Dentsply International Inc., York, PA, USA), a high-strength, water-absorbent dental molding material. Castone powder (120 g) was mixed with water (40 mL), and the slurry was then evenly divided among four empty dishes. Before the Castone could harden, dishes were gently and repeatedly tapped on a horizontal surface to ensure even distribution and remove air bubbles. After air-drying had occurred for ≈24 h, two holes (1.6 mm diam and 180° apart) were drilled through the side of the dish, just above the surface of the dried Castone, to provide entrance and exit holes for the ants. A third hole was drilled in the center of the accompanying lid of each dish. All dishes and lids were rinsed under running tap water to remove plastic debris and Castone dust prior to use. After rinsing, dishes and lids were placed in an oven (60 °C) for 1–3 d to ensure the complete drying of the Castone. After drying, and just prior to being used, the dishes were filled with water to ensure the complete saturation of the Castone.

### 2.3. Materials Evaluated as Deterrents to Argentine Ant Harboring

Laboratory trials were conducted to evaluate the deterrency of five materials to Argentine ant harboring behavior: fresh leaves from tansy plants (*Tanacetum vulgare* L.), spearmint plants (*Mentha* sp.), and rosemary plants (*Rosmarinus* sp.); the thin peel from a mature cucumber (*Cucumis* sp.); a water extract of soybean plants (*Glycine max* L.); and peppermint oil (positive control) and water (negative control) (Table 1). Peppermint oil is a strong deterrent to Argentine ant harboring [7]. It was acquired from Polarome International (Jersey City, NJ, USA) and formulated at 1% in n-hexane according to the method of Scocco et al. [7]. Tansy plants were purchased from a commercial nursery (Winterville, Georgia) and maintained in a greenhouse on the University of Georgia Griffin Campus. Spearmint and rosemary leaves were obtained from the garden of a local residence (Griffin, Georgia). The spearmint leaves were placed in a small plastic storage bag with water covering the leaves, while rosemary leaves were placed in a small plastic storage bag without water. Both spearmint and rosemary were used in bioassays within two hours of removal from the plant.

Cucumbers, purchased from a local supermarket, were washed thoroughly under tap water, and a thin layer of peel (≈1 mm thick) was removed (Farberware Euro Peeler, Lifetime Brands, Inc., Garden City, NY, USA). Round disks (≈3.5 mm diam) were prepared from cucumber peel, tansy leaf, and spearmint leaf using a standard paper hole punch (At the Office^TM^, Wal-Mart Stores, Inc., Bentonville, AR, USA).

A water extract from soybean plants, hereafter referred to as soybean tea, was prepared in a manner similar to that described by Bader [13]. In short, several soybean plants (Maturity Group 7, Roundup Ready, Georgia Crop Improvement Association, Inc., Athens, GA, USA) were harvested from the University of Georgia Griffin Campus research farm in Williamson, GA. Following their removal from the ground, the plants were placed in small, water-tight storage bags containing ≈250 mL of water. The plants remained in the open bag at ambient laboratory conditions until use (no more than 18 h). To prepare the soybean tea, the soybean plants were rinsed under tap water in order to remove dirt and debris. Leaves were removed from the plant, and 2.5 cm of the main stem was removed (the cut line was ≈6.5 mm above the first root). Two and a half cm of the remaining stem was then removed and cut into ≈6.5-mm sections, and the sections were then placed in a plastic vial (57 × 16.5 mm; Sarstedt Inc., Newton, NC, USA) containing 1 mL of tap water. The stems were allowed to soak for 24 h under ambient laboratory conditions, at which time they were removed, leaving behind the soybean tea used in our bioassays.

In the first trial, we evaluated the deterrent effect of four disks per replicate of dried cucumber peels and tansy leaves, fresh cucumber peels, tansy leaves, and spearmint leaves, and soybean tea (0.25 mL, or a 6.5-mm root equivalent). The controls consisted of 0.25 mL of 1% peppermint oil in n-hexane (positive control) and 0.25 mL of water (negative control) (Table 1). In the second trial, we evaluated 40 disks per replicate of fresh cucumber, fresh tansy, and fresh spearmint, four leaves of fresh rosemary (≈2.5 cm long each), and soybean tea (1.0 mL, or 26-mm root equivalents) (Table 1 and Figure 1). The controls consisted of 1.0 mL of 1% peppermint oil in n-hexane (positive control) and 1.0 mL of water (negative control).

### 2.4. Bioassay

In an approach similar to that of Scocco et al. [7], the treatment material or substance was applied directly to the surface of small (35 × 5 mm), round, plastic dishes half-filled with Castone; solid, leafy plant materials were placed inside the dishes in a manner that did not impede ant entry into the dish (Figure 1). For both trials, water (0.25 mL in trial 1 and 1.0 mL in trial 2) was first applied to the Castone to create an attractive, moist harborage for the ants [18]. For the soybean tea treatment, 0.25 mL (trial 1) or 1.0 mL (trial 2) was used in place of additional water. For the peppermint oil treatment, the solution was added to the stone dish and allowed to air-dry for 2 h in order to allow the hexane to completely evaporate (7); then, water was added. All liquid treatments were spread evenly over the Castone surface. Following the application of the treatment material and water, each dish was covered with its lid and placed in a small, plastic box (19 × 14 × 9.5 cm; Tri-State Plastics, Dixon, KY, USA) with Fluon-lined walls. Twenty worker ants were immediately added to each box. The worker ants used in all bioassays were collected from laboratory colonies by placing a dish containing ants (from the laboratory colony) into a small, Fluon-lined, plastic box (31 × 23 × 10 cm; Pioneer Plastics, Inc., Dixon, KY, USA) and allowing several ants to climb onto a small paintbrush. Twenty ants were then gently tapped into a clear, 30-mL plastic cup with the walls and floor coated with Fluon, and the ants were then gently transferred to test arenas containing a freshly treated dish.

During the bioassay, ants had the choice of entering the covered, moistened dish containing the treatment or not entering the dish. The number of ants inside the dish (alive + dead) was recorded after 2 and 4 h. Among all treatments, there was no mortality in trial one and negligible mortality (<1%) in trial two. All bioassays were conducted at room temperature, and each treatment in each trial was replicated 12 times (*n* = 12).

### 2.5. Statistical Analyses

For both trials, the treatment, the time, and the treatment x time interaction were analyzed using a mixed-model, two-way analysis of variance (ANOVA) (PROC GLIMMIX [19]). For each combination of trial and time, the mean number of live ants inside each treated harborage was analyzed using a mixed-model, one-way ANOVA (PROC GLIMMIX [19]). Following each one-way ANOVA, differences between least square means were determined using pairwise *t*-tests.

## 3. Results

In trial 1, the main effects, treatment (*F* = 18.43; df = 7, 176; *p* < 0.0001) and time (*F* = 4.97; df = 1, 176; *p* = 0.0271), were significant, but their interaction was not (*F* = 0.64; df = 7, 176; *p* = 0.7205). Excluding 1% peppermint oil, after 2 h, the mean number of ants inside the moistened, treated dishes was not significantly different (*F* = 8.73; df = 7, 88; *p* < 0.0001); the results were similar after 4 h (*F* = 14.56; df = 7, 88; *p* < 0.0001) (Table 2). Only the 1% peppermint oil treatment deterred ants from entering moistened dishes; none of the remaining treatments deterred the ants from entering dishes at either 2 or 4 h.

In the second trial, the main effects, treatment (*F* = 32.29; df = 6, 123.3; *p* < 0.0001) and time (*F* = 5.61; df = 1, 154; *p* = 0.0191), were significant, but their interaction was not (*F* = 0.88; df = 6, 123.3; *p* = 0.5085). Only fresh spearmint and fresh rosemary deterred ants from harboring; deterrence was defined as significantly fewer ants inside a harborage than in the water treatment (negative control). After 2 h, there were significantly fewer ants in dishes containing freshly collected spearmint and rosemary than in dishes containing only water (no treatment); dishes containing fresh rosemary contained fewer ants than dishes containing any treatment other than 1% peppermint oil (Table 2; *F* = 10.98; df = 6, 77; *p* <0.0001). Dishes containing spearmint contained significantly fewer ants than dishes containing soybean tea, but not fresh tansy or fresh cucumber. The results did not change appreciably after 4 h (*F* = 20.64; df = 6, 77; *p* < 0.0001). These results suggest that only fresh spearmint and fresh rosemary were deterrent to Argentine ant harboring and that fresh rosemary was a better deterrent than fresh spearmint.

## 4. Discussion

In the first trial, no treatment, excluding peppermint oil, which is a known Argentine ant deterrent, deterred ants from entering a preferred, moistened harborage; in each of the six treatments and water control, 88% to 97% of the ants entered and were harboring inside the harborage after 2 h, versus 93% to 99% after 4 h (Table 2). In the second trial, after 2 h, the water (negative control), soybean tea, and fresh tansy and cucumber had significantly similar responses; 68% to 92% of the ants had entered the treated harborages (Table 2). Fresh spearmint (47%), fresh rosemary (17%), and the peppermint oil solution (0.50%) were significantly different, but all demonstrated deterrency with ≤ 47% of the ants inside the treated harborages, and with the peppermint oil solution demonstrating a similar effect as that in trial 1. After 4 h, the ants’ response to the treatments had not changed significantly, but the response was greater, and more ants were inside the treated harborages (Table 2). The water control, soybean tea, and cucumber (93–96%) elicited similar responses among the ants inside the treated harborages. Using tansy led to 74% of the ants entering the treated harborages, versus 54% for spearmint. Fresh rosemary was a deterrent, and it led to 26% of ants entering the treated harborages. All treatments, with the exception of fresh spearmint, fresh rosemary, and the peppermint oil solution, had 74% to 96% of ants inside the dishes at the conclusion of the trial (after 4 h).

Increasing the concentration of fresh cucumber and fresh tansy 10-fold and soybean tea 4-fold (trial 2) did not improve the deterrent properties of these home remedies, which were not deterrent in either trial. Likewise, neither dried cucumber peel nor dried tansy leaves kept the ants from entering the moistened dishes (trial 1 only). The response of the ants to fresh spearmint, however, appeared to be concentration-dependent; in trial 1, the ants were not deterred by it, but when the concentration was increased ≈10-fold (trial 2), fresh spearmint deterred the ants from entering the treated dishes. In trial 1, after 4 h 96% of the ants had entered the dish containing spearmint, but when the concentration of spearmint was increased (trial 2), only 54% of the ants had entered the moistened dish containing fresh spearmint leaves after 4 h. Freshly picked rosemary leaves were also deterrent (trial 2). Among all the natural products tested, only freshly picked spearmint and rosemary leaves were deterrent to Argentine ants, and rosemary was a better deterrent than spearmint. None of the home remedies [13] deterred the ants from entering moistened harborages.

Essential oils represent one of four major types of botanical products in use for pest control, including pyrethrum, rotenone, and neem [14]. The recent, increased attention [8] given to plant essential oils as alternatives to traditional insecticides is largely due to their exemption from registration by the U.S. E.P.A. and their classification as so-called “25(b)” products, owing to the name of the federal act that granted their exemption from EPA registration [20,21]. According to Isman [14], this has caused an increase in the development and production of essential oil-based insecticides, fungicides, and herbicides for both commercial and residential use. Oils of particular interest are rosemary, cedar, clove, mint, and thyme [5,14,22,23]. Rosemary, for example, has fumigant, deterrent, and contact toxic properties against several insects, including stored product pests such as the bean weevil, *Acanthoscelides obtectus* (Say) [24,25]. Rosemary oil was toxic to adult turnip aphids, *Lipaphis pseudobrassicae* (Davis), the human head louse, *Pediculus humanus capitis* De Geer, the twospotted spider mite, *Tetranychus urticae* Koch, the armyworm, *Mythimna* (=*Pseudaletia*) *unipuncta* (Haworth), and the cabbage looper, *Trichoplusia ni* (Hübner) [25,26,27,28]. In addition to toxic characteristics, rosemary oil has shown promise as a viable deterrent to four mosquito species: *Anopheles stephensi* Liston, *Aedes aegypti* (L.), *Culex quinquefasciatus* Say, and *C. pipiens pallens* (L.) [29,30].

Argentine ants and red imported fire ants, *Solenopsis invicta* Buren, were given a choice of crossing an essential oil-treated paper bridge or a solvent-treated bridge to access food and water [6]. Basil, citronella, lemon, peppermint, and tea tree oils applied to the paper bridges at ≥0.40 μL per square centimeter deterred Argentine ant foraging. Foraging by red imported fire ants was deterred by citronella and peppermint (≥0.02 μL per square centimeter), basil and tea tree (≥0.40 μL per square centimeter), and lemon (≥2.0 uL per square centimeter). Neither ant species was deterred by eucalyptus at the rates tested (≤ 10.0 μL per square centimeter). In a lab study utilizing the same methods that we utilized in our study, fresh deposits (2 h old) of 0.10%, 1%, and 10% spearmint, peppermint, wintergreen, cinnamon, and clove oil deterred Argentine ants from their preferred harborage [7]. The deterrence appeared to be concentration-dependent. The same treatments were allowed to age (the top of the dish was removed to facilitate volatility) for one week, and the test was repeated. Only spearmint was deterrent at all three concentrations tested, while the other four essential oils (peppermint, wintergreen, cinnamon, and clove) retained their deterrent effects at the 1% and 10% concentrations.

Plant essential oils have been evaluated for their deterrent and insecticidal properties against *L. humile*; however, there have only been a limited number of laboratory tests [6,7] and limited field studies. A series of studies [4,5] investigated the impact of aromatic red cedar mulch on Argentine ant harboring in both laboratory and field trials. In a choice study where Argentine ants and Odorous house ants, *Tapinoma sessile* (Say), were provided the opportunity to nest in either cedar mulch or one of four other common mulches (pine straw, pine bark, shredded cypress, or chipped hardwood), both ant species always chose the non-cedar mulch (*n* = 78 trials) [4]. In the same study, Argentine ants suffered high mortality when confined to mulch (no choice) or when exposed to cedar mulch or cedar oil in a small, enclosed space. Clearly, cedar is both repellent and toxic (via fumigation and perhaps contact) to Argentine ants.

There was a strong, positive correlation between Argentine ant worker mortality and the amount of time the ants were in contact (i.e., the distance traveled) with cedar mulch, suggesting perhaps some contact mortality with cedar oil and/or its constituents. When forced to, Argentine ants forage over cedar mulch to reach food [5], but they prefer not to nest in it [4]. It seems the strong attraction to food may supersede the repellent nature of cedar, but not to the extent that the ants will nest in it. In a field trial in which large swaths of pine needle mulch were replaced with either cedar or cypress mulch, newly established Argentine ant nests were found in only three instances in the new cedar mulch but 26 times in cypress; and when pine straw mulch surrounding trees was replaced with either cedar or cypress, the number of nests found thereafter was significantly greater in cypress than in cedar [5].

The results from our natural product trials indicate that components of freshly harvested rosemary and spearmint were deterrent to Argentine ant harboring. In contrast, the home remedies evaluated in our study did not deter ants from a preferred harborage, even when the concentration of these remedies was increased 4- to 10-fold. The home remedies [13] we evaluated are unproven, and they are in direct comparison to products that have been vetted through a rigorous, data-driven, EPA registration process [31]. The U.S. Food and Drug Administration, likewise, administers a similar program for human drug development, whereby drug manufacturer claims are proven so that, when we take a drug, we indeed have confidence that the active ingredient has undergone a scientifically based, rigorous process that has led to evidence that the drug does indeed address a specific medical condition [32]. Because of U.S. federal government oversight, we are confident that the active ingredients in products designed to alter a pest’s biology or survival and to improve human health have been properly vetted through broadly accepted scientific guidelines.

We argue that the home remedies we have evaluated herein are placebos and produce a placebo-like effect among susceptible users. By definition, “a placebo is a pharmacologically inactive substance that can have a therapeutic effect if administered to a patient who believes that he or she is receiving an effective treatment” [33]. Bausell [33] goes on to state that “…the placebo effect is not something that occurs “naturally”. It must be manufactured in the sense that it occurs only in the presence of therapeutic intent (or the perception of such intent)”. The placebo effect has been widely studied since the 1950s, and it originates from medicine. Placebo effects are human responses to various external influences, and they are not unique to medicine. One of the most common influencers of the placebo effect is an authority figure (the physician in medicine). In our case, the authority figure is a book author and PhD who provides pest control recommendations—in our case, inert placebos [13]. We argue that there are close parallels, discussed below, between the belief in sham ant control recommendations (our study) and the general belief that a sugar pill cures ailments.

The literature surrounding placebos and the placebo effect originate from medicine, and it includes the existence of a placebo effect in such areas as pain remediation, psychiatry, urology, cardiology, and even surgery [34]. A growing body of medical researchers is investigating the biological nature of the placebo effect as a line of scholarly inquiry to gain insight into the function of the human brain [34] and the numerous mechanisms (e.g., the patient–physician relationship) that lead to a placebo effect.

In the proper context, a placebo treatment may alter brain chemistry, leading to slight improvements in a patient’s health—e.g., placebos (sugar pills) may indeed stimulate the brain to produce opioids as endogenous analgesics. Placebos may complicate the outcome of candidate drug efficacy trials, for instance, by stimulating various endogenous neuro-chemistries that might actually improve health outcomes in the absence of the candidate drug and thereby complicating the interpretation of the trial’s outcomes. This complicates studies designed to determine a candidate drug’s effectiveness, and it must, therefore, be controlled for by researchers in clinical research trials. Because the physician has such a strong influence on the patient’s view of the placebo, via enhanced expectations and beliefs, it is important that a placebo-controlled experiment be double blind—i.e., neither the patient nor the physician must know the identity of the placebo (sugar pill) or the candidate intervention being evaluated.

Although there are many mechanisms influencing the existence of a placebo effect, it is widely accepted that the doctor–patient relationship (i.e., a relationship with a trusted authority figure) greatly influences patients’ beliefs and expectations. Most of us have great respect for our doctors and other authority-type figures and are pre-disposed to trusting them. That, combined with the motivation to solve a pressing pest problem, may indeed lead to a placebo effect.

There are numerous other factors that may impact the occurrence of the placebo effect, and they include the following: the physical nature of the placebo itself—its presentation and look, including packaging and perhaps color; the so-called “halo effect”, referring to the product’s reputation; and even the price of the product [34]. Other important mechanisms enhancing the placebo effect are patient-dependent, and they include a person’s personality profile (optimistic people are more susceptible to the placebo effect), the patient’s positive expectation toward the “treatment”, and their belief or faith in its performance. A placebo may result in a positive outcome if the patient expects it to, and this expectation is influenced (positively and negatively) by the physician (a trusted authority figure) [35].

Individuals seeking relief from infestations of insects are motivated to find a cure for their ailment—in this case, infesting ants, and they are pre-disposed to intent from an authority figure. Pest control devices emitting ultrasonic sound waves have been sold for decades, and they can still be found on store shelves today, even though numerous studies have proven their ineffectiveness against repelling or killing common pests of households. Hinkle et al. [36], for instance, affixed ultrasonic collars to cat flea (*Ctenocephalides felis* Bouchë)-infested house cats to evaluate the device’s impact on several vital rates of the fleas. The ultrasonic devices had no impact on egg production by female fleas, larval development time, pupa production, or adult survival. Similar studies have shown the lack of detrimental impacts of ultrasonic devices on German cockroaches, *Blattella germanica* (L.) [37].

In conclusion, the recommended home remedies evaluated in our trials were not deterrent to Argentine ant harboring, and it is our opinion that users of remedies that have not been scientifically vetted are susceptible to these recommendations in a manner similar to the response to placebos, through which an authority figure (in medicine, the physician) greatly influences, positively and negatively, user expectations and beliefs. Moreover, our results suggest that a further investigation, in the form of field trials, of rosemary and spearmint is warranted and might provide valuable insight into effective plant-based management for *L. humile* [5]. Given the consistent interest in alternatives to conventional insecticides, it is not surprising that natural remedies have been suggested to meet this demand. We believe the application of granular formulations of essential oils to areas where pest ants nest might serve to create an ant-free zone around susceptible structures [5]. Essential oils have their greatest utility as deterrents, but because of their high volatility, repeat applications might be necessary. Because of their deterrent nature, essential oils should primarily be considered as behavior-modifying chemicals and secondarily as contact toxicants.

## Figures and Tables

**Figure 1 insects-15-00768-f001:**
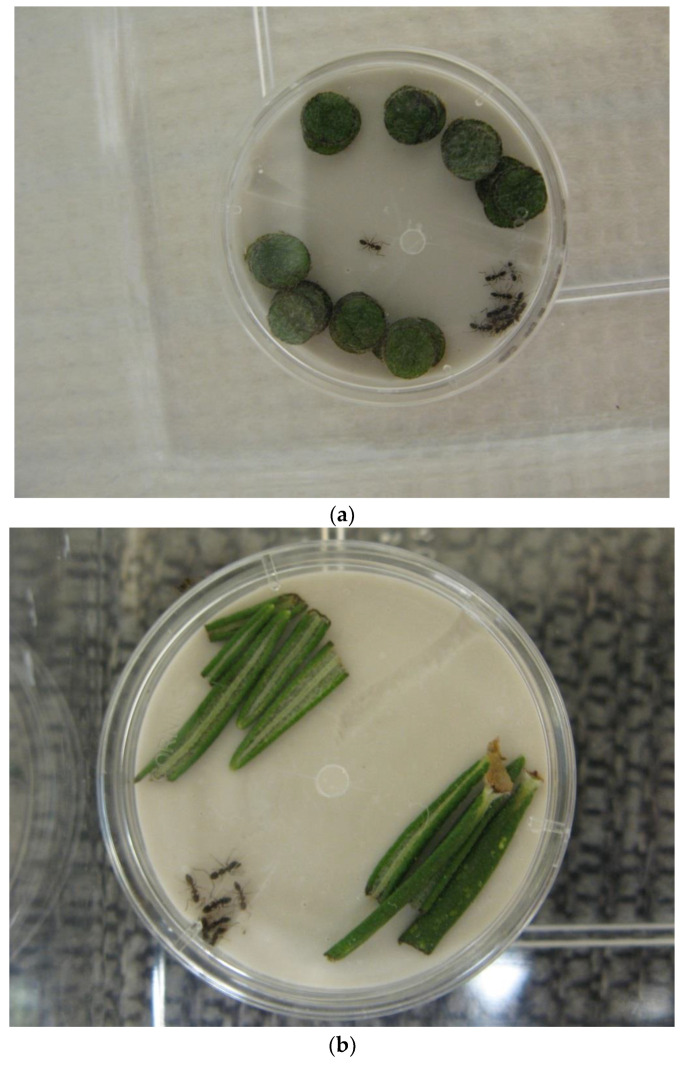
Test arena (35 mm-diameter dish half-filled with water-absorbent Castone and with entry holes) containing (**a**) 40 disks of fresh spearmint leaves (trial 2) and (**b**) freshly cut leaves from a rosemary plant (trial 2). In a and b, note that the ants inside the dishes are congregated next to one of the entry holes, as fresh cuttings from both plants were deterrent to the ants.

**Table 1 insects-15-00768-t001:** Quantity of treatment material applied to harborage for natural product deterrency trials.

	Amount of Material or Substance Tested for Each of Two Trials	
Treatment	Trial 1 *	Trial 2 **
Fresh cucumber Dry cucumber ^a^ Fresh tansy Dry tansy ^a^ Soybean tea Fresh spearmint Fresh rosemary Water only (negative control) 1% peppermint oil (positive control)	4 disks (11.3 cm^2^) 4 disks (11.3 cm^2^) 4 disks (11.3 cm^2^) 4 disks (11.3 cm^2^) 0.25 mL (6.5 mm RE ^b^) 4 disks (11.3 cm^2^) Not tested 0.25 mL 0.25 mL	40 disks (113 cm^2^) Not tested 40 disks (113 cm^2^) Not tested 1.0 mL (26 mm RE ^b^) 40 disks (113 cm^2^) Whole leaves 1.0 mL 1.0 mL

* For trial 1, 0.25 mL of water was added to all dishes just prior to the addition of the candidate treatment; for the soybean tea treatment, 0.25 mL of the tea was added. ** For trial 2, 1.0 mL of water was added to all dishes just prior to the addition of the candidate treatment; for the soybean tea treatment, 1.0 mL of the tea was added. ^a^ Dried for 24 h at 40 °C. ^b^ RE, root equivalent. RE is the equivalent length of stem from a soybean plant.

**Table 2 insects-15-00768-t002:** Response of Argentine ants to various treatments applied to harborages for natural product harborage deterrency trials.

	Number (mean ± S.E.) of Live Ants Inside Dish at Hour *			
	Trial 1		Trial 2	
Treatment	2 h	4 h	2 h	4 h
Dry cucumber Fresh cucumber Dry tansy Fresh tansy Soybean tea Fresh spearmint Fresh rosemary Water-only control 1% peppermint oil	19.4 ± 1.3 A (97) 17.6 ± 1.2 A (88) 18.9 ± 1.3 A (95) 18.4 ± 1.2 A (92) 19.3 ± 1.3 A (97) 17.6 ± 1.2 A (88) --- 18.4 ± 1.2 A (92) 0.28 ± 0.2 B (1)	19.8 ± 1.3 A (99) 18.5 ± 1.2 A (93) 18.7 ± 1.2 A (94) 19.6 ± 1.3 A (98) 19.4 ± 1.3 A (97) 19.2 ± 1.3 A (96) --- 19.1 ± 1.3 A (96) 0.92 ± 0.3 B (5)	--- 13.6 ± 2.5 A,B (68) --- 14.8 ± 2.7 A,B (74) 16.8 ± 3.1 A (84) 9.33 ± 1.8 B (47) 3.42 ± 0.8 C (17) 18.3 ± 3.3 A (92) 0.08 ± 0.09 D (0.5)	--- 18.5 ± 2.2 A (93) --- 14.8 ± 1.9 A,B (74) 19.2 ± 2.3 A (96) 10.7 ± 1.4 B (54) 5.17 ± 0.8 C (26) 19.1 ± 2.3 A (96) 0.50 ± 0.2 D (3)
	*F* = 8.73 df = 7, 88 *p* < 0.0001	*F* = 14.56 df = 7, 88 *p* < 0.0001	*F* = 10.98 df = 6, 77 *p* < 0.0001	*F* = 20.64 df = 6, 77 *p* < 0.0001

Following a mixed-model, one-way ANOVA, differences between least square means for each combination of trial and hour were determined using pairwise *t*-tests; means within a column followed by the same letter are not significantly different. * Numbers in parentheses are the average percentage (*n* = 12) of 20 ants found inside the dish.

## Data Availability

Contact D.R.S. for acquisition of raw data.

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
