# Peer review of "Common Home Remedies Do Not Deter Argentine Ants, Linepithema humile (Hymenoptera: Formicidae), from a Preferred Harborage"

_insects, 2024, doi:10.3390/insects15100768_

Round 1

Reviewer 1 Report

Comments and Suggestions for Authors

In this study, several items were evaluated as potential shelter deterrents for the Argentine ant.

In two simple experiments with a well-designed set up, the authors measured whether ants entered the damp shelter or whether the items placed inside deterred ants from entering. Appropriate positive and negative controls were used.

In an era when social media often blurs the line between what is true and what is not, it is important to be able to discard unfounded claims. This simple study contributes to that goal.

General concerns:

Some of the items tested were chosen based on their recommendation in a (non-scientific) book as home remedies against ants. For other elements, there is no explanation for their selection; they may have been chosen based on information circulating on social media. Please provide an explanation for why these items were chosen.

Two concentrations and two time points were evaluated: 2 and 4 hours. It would be helpful to understand why only these time points were selected, as this appears to be a limitation of the study design. To determine whether any item can effectively deter ants, it would be advisable to test for longer periods than 4 hours, especially if considering the item as a potential method for ant management.

Some issues that are not directly relevant to the study’s focus are discussed at length, while other important aspects that could have been addressed are omitted (see minor concerns).

The table presents the results but does not include contrast statistics. It would be helpful to see these results in a more accessible format. The authors could create two similar figures for each test, with each figure containing two panels: one for 2 hours and one for 4 hours. This would provide a clearer and quicker view of the results.

Minor concerns:

Line 154: I assume that the time mentioned is sufficient for the solvent to evaporate. If so, please state this, and if there is a reference where this was tested, please include it.

Where were the leaves placed inside the dish? Was care taken not to place them too close to the entrance holes? Were they placed symmetrically on both sides, as shown in the photos? When there were only 4 discs, were they stacked, or separated and distributed on both sides? How far were they from the entrance holes? All of this information could affect ant entry.

Line 173-174: Include whether the assumptions of the model are met, such as the distribution of residuals, etc. Was 'replica' included as a random explanatory variable?

Line 188: You say: “Numbers in parentheses are the percentage of 20 ants found inside the dish”. 

Do the numbers in parentheses represent the mean of 12 replicates for the percentage of 20 ants found inside the dish?

It is reasonable to assume that works lacking scientific rigor could result in subjective perceptions of the effects of certain substances. You mention the possibility of producing a placebo-like effect in susceptible users, but you only explicitly explain what that effect might be in this case, in line 336. Instead, from lines 294 to 335, you focus on the placebo effect in medicine and pharmacology. This entire discussion about placebos in medicine should be removed, as it does not contribute to the overall work.

I recommend providing a convincing explanation for why the study did not extend beyond 4 hours (e.g. 8, 12, and/or 24 hours). Discuss the importance of the effect lasting for a sufficient duration to be practical for ant management.

Author Response

Comments and Suggestions for Authors

In this study, several items were evaluated as potential shelter deterrents for the Argentine ant.

In two simple experiments with a well-designed set up, the authors measured whether ants entered the damp shelter or whether the items placed inside deterred ants from entering. Appropriate positive and negative controls were used.

In an era when social media often blurs the line between what is true and what is not, it is important to be able to discard unfounded claims. This simple study contributes to that goal.

General concerns:

Some of the items tested were chosen based on their recommendation in a (non-scientific) book as home remedies against ants. For other elements, there is no explanation for their selection; they may have been chosen based on information circulating on social media. Please provide an explanation for why these items were chosen. Change made. Inserted this text in the Introduction: we also evaluated the deterrent nature of freshly-harvested leafy material from spearmint and rosemary plants, as commercially available essential oil extracts from these plants are known deterrents (6,7).

Two concentrations and two time points were evaluated: 2 and 4 hours. It would be helpful to understand why only these time points were selected, as this appears to be a limitation of the study design. To determine whether any item can effectively deter ants, it would be advisable to test for longer periods than 4 hours, especially if considering the item as a potential method for ant management. We respectfully disagree with this point. The positive and negative controls in our study controlled for this quite nicely. If a substance is not deterrent to ants immediately (2 h), it will not become deterrent with time (>4 h) (see refs 6 and 7). Deterrent effects are immediate. Alternatively, when a substance is immediately deterrent it may lose its deterrent nature with time/aging (see reference 7). In the positive control (1% peppermint oil) in both trials less <1% of the ants (12 replicates of 20 ants) had entered the dish after 2 hours, and 5% or less after 4 hours. In the negative control (water), in both trials 92% of the ants had entered the dish at 2 h and 96% at 4 h. The positive and negative controls acted exactly as they should have, thereby providing confidence in the results we see from each of the assigned treatments. The controls behaved as they should have.

Some issues that are not directly relevant to the study’s focus are discussed at length, while other important aspects that could have been addressed are omitted (see minor concerns).

The table presents the results but does not include contrast statistics. It would be helpful to see these results in a more accessible format. The authors could create two similar figures for each test, with each figure containing two panels: one for 2 hours and one for 4 hours. This would provide a clearer and quicker view of the results. Data for both trials were analyzed as a straightforward two-way analysis of variance: treatment time treatment x time interaction. In neither trial was the interaction significant (lines 182 and 195), Statistically, then, it is legal to hold one variable constant (time) while analyzing (via one-way ANOVA) the effect of treatment. The results of these one-way analyses are reported at the bottom of Table 2 (F, df, P-value). We believe these date are well-represented in tabular form, especially with letter assignments showing significant differences as a result of all pairwise comparisons. We don’t believe a graph would add much to the results.

Minor concerns:

Line 154: I assume that the time mentioned is sufficient for the solvent to evaporate. If so, please state this, and if there is a reference where this was tested, please include it. We added: “to allow the hexane to completely evaporate (7)”.

Where were the leaves placed inside the dish? Was care taken not to place them too close to the entrance holes? Were they placed symmetrically on both sides, as shown in the photos? When there were only 4 discs, were they stacked, or separated and distributed on both sides? How far were they from the entrance holes? All of this information could affect ant entry. We added: ; solid, leafy plant materials were placed inside the dishes in a manner that did not impede ant entry into the dish.

Line 173-174: Include whether the assumptions of the model are met, such as the distribution of residuals, etc. Was 'replica' included as a random explanatory variable? The SAS procedure GLIMMIX controls for any violation of normality, and rep was a random variable.

Line 188: You say: “Numbers in parentheses are the percentage of 20 ants found inside the dish”. Do the numbers in parentheses represent the mean of 12 replicates for the percentage of 20 ants found inside the dish? Yes. Clarified in Table 2 by addition of word “average” (before the word percentage) and “(n = 12)” (after the word percentage).

It is reasonable to assume that works lacking scientific rigor could result in subjective perceptions of the effects of certain substances. You mention the possibility of producing a placebo-like effect in susceptible users, but you only explicitly explain what that effect might be in this case, in line 336. Instead, from lines 294 to 335, you focus on the placebo effect in medicine and pharmacology. This entire discussion about placebos in medicine should be removed, as it does not contribute to the overall work. We have removed some language referring to the placebo effect and tightened up our argument to make it more clear. We argue that these home remedies for ant control are sham recommendations (placebos), and that people who are “fooled” by them are susceptible to their effectiveness, much the same that a sugar pill (placebo, in medicine) elicits positive responses in susceptible individuals. The literature surrounding the placebo effect is unique and has its origins in medicine, and our argument is best made (as a potential influence) by comparison to the rich literature documenting the placebo effect in medicine. By explaining the placebo effect in medicine we are attempting to explain the attractiveness and perceived positive effects of unvetted, sham recommendations for pest control. There are many parallels. We must explain what a placebo effect is in order to shed light on our argument and expose the parallels.

The placebo effect (in medicine) has been widely studied since the 1950s and has its origins in medicine, but that doesn’t make placebo-like effects unique to medicine. Placebo effects are human responses to various external influences. Placebo effects are ubiquitous, it’s just that medicine has formally identified influencing variables (such as an authority figure with a Ph.D. after his name and a book selling sham pest control recommendations! [see ref 13]) that allow the placebo effect to be expressed. We argue that there are many close parallels between belief in sham ant control recommendations (our study) and the belief that a sugar pill cures all ailments. And we make that argument by explaining common aspects of the placebo effect in medicine. The Discussion section of a manuscript is the exact place for these conversations to be had. It’s up to the reader to decide the validity of our argument, and they are free to agree or disagree with what we’ve proposed.

I recommend providing a convincing explanation for why the study did not extend beyond 4 hours (e.g. 8, 12, and/or 24 hours). Discuss the importance of the effect lasting for a sufficient duration to be practical for ant management. Repeat of Answer Above: The positive and negative controls in our study controlled for this quite nicely. If a substance is not deterrent to ants immediately (2 or 4 h), it will not become deterrent with time (>4 h). Deterrent effects are immediate. Alternatively, when a substance is immediately deterrent it can lose its deterrent nature with time/aging (see reference 7). In the positive control (1% peppermint oil) in both trials less <1% of the ants (12 replicates of 20 ants) had entered the dish after 2 hours, and 5% or less after 4 hours. In the negative control (water), in both trials 92% of the ants had entered the dish at 2 h and 96% at 4 h. The positive and negative controls acted exactly as they should have, thereby providing confidence in the results we see from each of the assigned treatments. The controls behaved as they should have.

Reviewer 2 Report

Comments and Suggestions for Authors

In this work, the authors evaluated the repellent effect of several common household treatments (cucumber peel, salvia leaf, soya bean extract, spearmint leaf and rosemary leaf) on ants by designing two simple experiments and found that most of these remedies had no substantial effect on argentine ants. The study is interesting and provides a foundation for the development of botanical insecticides. I would like to suggest this manuscript to be accepted for publication in your esteemed journal "insects". But there are several questions on writing and experiment design. A few minor suggestions should be considered before publication. The major comments are as follows:

1.     As required by the journal, graphical abstract is also required from the authors.

2.     Line 43, It is suggested that additional relevant background on Argentina ant infestation.

3.     Line 64. Are ants used for bioassays uniform in size? This should be stated in the ant presentation.

4.     In Table 2, I would suggest that the percentages be listed in a separate column so that it looks clearer.

5.     Line 98 and 255, peppermint, wintergreen, cinnamon, and clove oil  can all deterred Argentine ants from their preferred, so why was peppermint oil chosen as the positive control.

6.     Is there a basis for the dosage of soybean tea, and are higher concentrations of soybean t tea effective?

7.     The fresh rosemary leaves have been effective against ants, however, in the experimental design, fresh rosemary leaves were only present in trial 2, why was trial 1 not used, the authors should have given clarification or added this trial.

8.     Line 171, “treatment*timesuggested should be revised to “treatment Í time

9.     Line 185. one-way ANOVA differences.

10.  Line 205-231. I don't think this section belongs in the discussion and would be more appropriately placed in the results section.

Line 262, [6,7[should be revised to“[6,7]”

Author Response

Comments and Suggestions for Authors

In this work, the authors evaluated the repellent effect of several common household treatments (cucumber peel, salvia leaf, soya bean extract, spearmint leaf and rosemary leaf) on ants by designing two simple experiments and found that most of these remedies had no substantial effect on argentine ants. The study is interesting and provides a foundation for the development of botanical insecticides. I would like to suggest this manuscript to be accepted for publication in your esteemed journal "insects". But there are several questions on writing and experiment design. A few minor suggestions should be considered before publication. The major comments are as follows:

  1. As required by the journal, graphical abstract is also required from the authors. We have submitted an outstanding, close-up photograph of Argentine ant, Linepithema humile, workers.
  2. Line 43, It is suggested that additional relevant background on Argentina ant infestation. On line 41 we state that the Argentine ant is a “major nuisance pest”.
  3. Line 64. Are ants used for bioassays uniform in size? This should be stated in the ant presentation. On line 42 we inserted language to describe the ant: “small (3 mm), monomorphic”
  4. In Table 2, I would suggest that the percentages be listed in a separate column so that it looks clearer. In our opinion separating the percentages into additional columns in the table makes the table cluttered.
  5. Line 98 and 255, peppermint, wintergreen, cinnamon, and clove oil  can all deterred Argentine ants from their preferred, so why was peppermint oil chosen as the positive control. We chose 1% peppermint oil as a positive control because we know it is highly deterrent to Argentine ant harboring (see reference 7). See line 99.
  6. Is there a basis for the dosage of soybean tea, and are higher concentrations of soybean t tea effective? There is no basis for the concentration tested. The recommendation from Bader (see reference 13) simply suggested making a “soybean tea” from soybean plants. We followed his methods on how to prepare the tea, and then increased the concentration 4-fold to fairly assess the extract. Bader provides just vague methodology.
  7. The fresh rosemary leaves have been effective against ants, however, in the experimental design, fresh rosemary leaves were only present in trial 2, why was trial 1 not used, the authors should have given clarification or added this trial. The deterrent impacts of freshly-picked rosemary leaves were assessed in trial 2, but not trial 1, simply because of timing. Since we were assessing freshly-picked spearmint leaves we added rosemary leaves as an additional treatment because prior studies (references 6,7) showed myriad plant based essential oils to be deterrent.to Argentine ant harboring. As clarification, we inserted the following text into the Introduction: “we also evaluated the deterrent nature of freshly-harvested leafy material from spearmint and rosemary plants, as commercially available essential oil extracts from these plants are known deterrents (6,7)”.
  8. Line 171, “treatment*time”suggested should be revised to “treatment Í time” Changed the asterisk to an “x”. so it now reads “treatment x time”. I could not find the symbol suggested.
  9. Line 185. one-way ANOVA differences. Change made.
  10. Line 205-231. I don't think this section belongs in the discussion and would be more appropriately placed in the results section. We respectfully disagree with this. We believe that the Results section of manuscripts are where the statistical outcomes of a study are delivered, and nothing else. The Discussion section, however, is the location where the results can be discussed freely, without hindrance by statistical statements. That is what we’ve done on lines 205-231.

Line 262, “[6,7[”should be revised to“[6,7]” Change made.

Reviewer 3 Report

Comments and Suggestions for Authors

The manuscript titled. “Common Home Remedies Do Not Deter Argentine Ants, Linepithema humile (Hymenoptera: Formicidae), From a Preferred Harborage” addresses a rather necessary topic concerning the experimental testing of ant repellent properties of some natural plant products commonly used in households. There is a growing interest in these products in the US, as they tend to be safe in this regard, unlike the usual synthetic substances that are toxic to humans and the environment. Overall, the work has average scientific value, contains a number of ambiguities and needs to be restructured, especially in the Discussion. The chapter should be more clearly written and limited to specific, concerning strictly the research topic addressed (about the details below in the commentary). After clarification and completion according to my suggestions, possible publication of this manuscript can be considered, but I leave the final decision on this issue to the Editor of Insects.

My comments and suggestions are as follows: 

1. The title of the paper is not entirely appropriate to its content. According to the analysis, of the five natural products tested, two of them - fresh green mint leaves and fresh rosemary leaves (to varying degrees) nevertheless repelled ants in the second trial. Unless, these two natural products are not on Bader's (2007) list. If so, why were they studied by the authors in the paper with the above title. Also, did the authors study all the commonly known, home remedies against  ants? If only selected ones, they should not make the title of the paper so generalized.

2. Why exactly these natural products were chosen for the study, what was the guiding principle? Why such a variety of their types (e.g., leaves, aqueous extract, cucumber peels, oil), where the availability of active substances to repel ants is very different (e.g., leaves and oil), were selected and compared.  Wouldn't it be better to test only one type of natural product, for example, an aqueous extract obtained from the all plants used for testing?

3. Has information on the effectiveness of tested natural ant repellent products been described only in the work of Bader (2007)? It would have been useful in the Introduction to describe the history of the use of these products by humans, from which places associated with humans ants were repelled in the past. Which ant species these products were supposed to repel? Perhaps other ant species, however, react differently to the natural products studied than the Argentine ant. 

1.    4. Comparing in the Discussion people's use of scientifically unproven natural ant repellents to the medical placebo effect is cursory and unauthorized. The purchase and use of unproven natural ant repellents, despite their questionable effectiveness, occurs as a result of classic social engineering, pushy advertising and manipulation of people. As is the case with the purchase of other purchased goods. The placebo effect is a phenomenon in the history of medicine that has been scientifically documented many times, although its mechanism is not yet thoroughly studied. In its case, however, there is always a beneficial!!! for a patient a therapeutic effect, formed as a result of positive mental stimulation, despite the lack of classical action of the administered drug. This is quite different from the case of some people's unreflective use of natural ant repellents, with questionable effects. Another thing is that the authors do not give in the paper the scale of this phenomenon in the United States. I also do not see the point of such a broad description of the placebo effect itself in the Discussion. After all, the topic of the paper is about something completely different.

Author Response

The manuscript titled. “Common Home Remedies Do Not Deter Argentine Ants, Linepithema humile (Hymenoptera: Formicidae), From a Preferred Harborage” addresses a rather necessary topic concerning the experimental testing of ant repellent properties of some natural plant products commonly used in households. There is a growing interest in these products in the US, as they tend to be safe in this regard, unlike the usual synthetic substances that are toxic to humans and the environment. Overall, the work has average scientific value, contains a number of ambiguities and needs to be restructured, especially in the Discussion. The chapter should be more clearly written and limited to specific, concerning strictly the research topic addressed (about the details below in the commentary). After clarification and completion according to my suggestions, possible publication of this manuscript can be considered, but I leave the final decision on this issue to the Editor of Insects.

My comments and suggestions are as follows:

  1. The title of the paper is not entirely appropriate to its content. We disagree. The title of the paper is exclamatory and describes exactly the findings of the study.
  2. According to the analysis, of the five natural products tested, two of them - fresh green mint leaves and fresh rosemary leaves (to varying degrees) nevertheless repelled ants in the second trial. Unless, these two natural products are not on Bader's (2007) list. If so, why were they studied by the authors in the paper with the above title. Prior studies (references 6,7) showed myriad plant based essential oils to be deterrent.to Argentine ant harboring. We evaluated the raw material (freshly-picked leaves) to confirm the deterrency of these secondary plant compounds in their natural state (plant leaves). Inserted this text in the Introduction: “we also evaluated the deterrent nature of freshly-harvested leafy material from spearmint and rosemary plants, as commercially available essential oil extracts from these plants are known deterrents (6,7)”.
  3. Also, did the authors study all the commonly known, home remedies against  ants? If only selected ones, they should not make the title of the paper so generalized. No, we did not assess all home remedies. There are multiple dozens, if not hundreds, of unproven home remedies proposed by lay people for the control of pests. For example, the title of Bader’s book is titled “1001 All Natural Secrets…”. The scientific community has rarely provided an accurate assessment of these myriad claims, leaving the public with not knowing what to believe. The intent of this study was to demonstrate the need to trust only properly-vetted products and active ingredients that have been through a rigorous scientific process, and to think critically about all future claims and recommendations and the source of the information.
  4. Why exactly these natural products were chosen for the study, what was the guiding principle? Why such a variety of their types (e.g., leaves, aqueous extract, cucumber peels, oil), where the availability of active substances to repel ants is very different (e.g., leaves and oil), were selected and compared.  Wouldn't it be better to test only one type of natural product, for example, an aqueous extract obtained from the all plants used for testing? Again, the intent of the study was to assess, in a controlled environment, extraordinary claims (underpinned by no supportive data) put forth in a widely-available reference claiming to provide efficacious, easily constructed methods for controlling nuisance ants. The Argentine ant is a widespread nuisance ant pest in North America, and was a good model species for this study. We assessed the claims by following (exactly) the methods put forth by this reference book (reference 13).
  5. Has information on the effectiveness of tested natural ant repellent products been described only in the work of Bader (2007)? There is a large literature on the toxicity and repellency of essential oils on insect pests, and we cite many of those studies in our MS. The type of work recommended by Bader, however, has nothing published mainly because these recommendations do not come from a place where efficacy is important.
  6. It would have been useful in the Introduction to describe the history of the use of these products by humans, from which places associated with humans ants were repelled in the past. Which ant species these products were supposed to repel? Perhaps other ant species, however, react differently to the natural products studied than the Argentine ant. The Argentine ant is invasive to North America, and has spread globally over the past 100 years. The justification for using this ant in our studies is two-fold - it is a major nuisance pest and is therefore an ideal model ant for these assessments. Perhaps our study will spark future studies by authors to assess these treatments, and additional unvetted, home remedies, on additional pest ants.
  7. Comparing in the Discussion people's use of scientifically unproven natural ant repellents to the medical placebo effect is cursory and unauthorized. The purchase and use of unproven natural ant repellents, despite their questionable effectiveness, occurs as a result of classic social engineering, pushy advertising and manipulation of people. As is the case with the purchase of other purchased goods. The placebo effect is a phenomenon in the history of medicine that has been scientifically documented many times, although its mechanism is not yet thoroughly studied. In its case, however, there is always a beneficial!!! for a patient a therapeutic effect, formed as a result of positive mental stimulation, despite the lack of classical action of the administered drug. This is quite different from the case of some people's unreflective use of natural ant repellents, with questionable effects. Another thing is that the authors do not give in the paper the scale of this phenomenon in the United States. I also do not see the point of such a broad description of the placebo effect itself in the Discussion. After all, the topic of the paper is about something completely different. We disagree. The placebo effect has been widely studied since the 1950s and has its origins in medicine, but that doesn’t make placebo-like effects unique to medicine. Placebo effects are ubiquitous, it’s just that medicine has formally identified influencing variables (such as an authority figure with a Ph.D. after his name and a book selling sham pest control recommendations! [see ref 13]) that allow the placebo effect to be expressed. We argue that there are many close parallels between belief in sham ant control recommendations (our study) and the belief that a sugar pill cures all ailments. And we make that argument by explaining common aspects of the placebo effect in medicine. The Discussion section of a manuscript is the exact place for these conversations to be had. It’s up to the reader to decide the validity of our argument, and they are free to agree or disagree with what we’ve proposed.

Round 2

Reviewer 2 Report

Comments and Suggestions for Authors

no

Reviewer 3 Report

Comments and Suggestions for Authors

The manuscript can be published in the submitted version.